# OpenReview forum: "Linearly Decoding Refused Knowledge in Aligned Language Models"
_ICLR.cc/2026/Conference — ICLR 2026 Conference Withdrawn Submission_

### Official Review · Reviewer_GtJt · 2025-10-21

**Soundness:** 3
**Presentation:** 3
**Contribution:** 1
**Rating:** 2
**Confidence:** 4

**Summary:**

The paper performs a number of experiments predicting "harmful responses" from internal representations of LLMs. These harmful responses are scaler quantities that most LLM with safety fine tuning will refuse to generate. The internal representations are residual stream are a generic statements about the entity in question. The experiments show a linear model can predict these harmful responses with reasonable accuracy from these internal representations. There linear predictions some times persist in performance during fine tuning, and can be predictive of how the model will respond to pair wise comparison questions.

**Strengths:**

The paper is mostly easy to follow. It introduces as simple idea and performs experiments based on that. It also does a good job explaining the relation to prior work.

**Weaknesses:**

While I do not find anything in the paper factually wrong, the contribution and scope of experiments are very limited to the point where I think the paper likely requires additional content (experiments) or would be better suited to a workshop. The paper presents very little in the sense of knew methods and is more a understanding paper. I have no issue with this sort of submission, but they are a lot more compelling when they present results that are surprising or interesting. The fact that information is recoverable from linear probes, is consistent given prior work on how refusal mechanically manifests in LLMs, or as the paper puts it "Superficial Alignment". To the best of my knowledge this is the dominate paradigm so a few results that a inline with it just isn't that interesting or significant a contribution. In the Conclusion the authors state "Moreover, the decoded attributes correlate with model behaviour in comparative tasks, hinting at the notion that models may be “using” these representations." I think this is telling that the author them selves believe the result as they stand are likely insufficient to do more than hint. I really encourage expanding the experiments and resubmitting when more conclusive statements can be made.

Going into more detail the weaknesses of the paper are as follows:
- The presentation of the results could be improved there are only three plots in the main body of the paper, showing only one of the entity types, I'm sure plots showing more results, or more plot could be presented in the main paper. The text in the main paper reference plots in the appendix a lots this is fine in small doses but the current presentation is awkward. I'd suggest the authors tried to make the last two pages less verbose and include more plots in the main body.
- The discussion section, is long not very precise or well written, for example the 3rd paragraph is nearly all one long sentence.
- No mention of unlearning. The goal of refusal training is not to remove knowledge from the network so it can't be recovered, that is the field of unlearning and hence some mention of unlearning or how the work relates to techniques used in unlearning would seem wise. If model providers really wanted to ensure this sort of information was not recoverable unlearning techniques would be used rather than refusal training.
- Only three models investigated from two model families
- Only predicting scalar harmful responses
- Only using two jail break prompts
- Only using linear probes
- In general the experiments are just very limited, while not seeming particularly expensive to run.
- The fact the paper only consider scaler quantises could be made more clear in the abstract/introduction

**Questions:**

If you were going to perform more experiments what would they be?

Can you comment on the relation of your work to unlearning?

---

### Official Review · Reviewer_NtjF · 2025-10-31

**Soundness:** 2
**Presentation:** 3
**Contribution:** 2
**Rating:** 4
**Confidence:** 3

**Summary:**

This paper investigates the extent to which information that aligned language models (LMs) refuse to generate in response to user prompts remains accessible in their hidden representations. The authors use linear probing approaches across several instruction-tuned LMs (gemma-2-9b-it, gemma-2-2b-it, Yi-6B-Chat) to decode values for controversial or refused queries (e.g., average IQ for a country), using both innocuous and jailbreak prompts. The experiments demonstrate that a significant portion of refused knowledge is linearly decodable from hidden states, and that probes trained on base (non-refusing) models can often transfer to instruction-tuned counterparts. Furthermore, the decoded representations align with downstream behaviors, such as pairwise comparison tasks, suggesting that refused knowledge can still subtly influence model outputs despite alignment procedures.

**Strengths:**

1. This research investigates a critical question in AI alignment and interpretability: whether knowledge that a large language model is trained to refuse remains accessible after safety tuning.

2. The research is supported by a thorough and transparent experimental design, featuring a clear methodology for training linear probes, meticulous documentation of prompt engineering, and comprehensive validation procedures.

**Weaknesses:**

1.	The analysis is strictly empirical and would be strengthened by a theoretical or mechanistic explanation for the observed phenomena.
2.	The research relies on manually constructed data, which limits the generalizability of its findings to real-world LLM alignment and jailbreaking scenarios.
3.	The use of linear probes is a significant limitation, as this method is ineffective in common, practical scenarios where data is not linearly separable.

**Questions:**

1.	The analysis relies on linear probes to measure accessible knowledge. What specific advantages did this method offer for your research question? Furthermore, could an alternative approach, like training a small decoder on multiple hidden layers, provide a more robust measure by capturing non-linear relationships in the model’s representations?
2.	The results in Figure 3 indicate that probe transfer success varies substantially depending on the entity type. Could you expand on the underlying mechanisms that might explain this variance?
3.	Large Language Models produce probabilistic outputs, meaning different answers can be sampled from the same prompt. How is this inherent uncertainty incorporated into your definition and measurement of ‘accessible information’? For example, does your framework model the full output distribution, or does it rely on a single, most-likely generation?

---

### Official Review · Reviewer_b5Gv · 2025-10-31

**Soundness:** 3
**Presentation:** 3
**Contribution:** 2
**Rating:** 4
**Confidence:** 4

**Summary:**

This paper investigates how aligned language models internally represent knowledge they refuse to express directly. The authors demonstrate that: (1) refused information is linearly decodable from hidden states using probes trained on inputs, (2) these representations persist from base models through instruction-tuning, and (3) the decoded information influences downstream behavior in implicit decision-making tasks. The findings suggest current alignment methods suppress expression of undesirable content without eliminating the underlying representations.

**Strengths:**

1. The paper introduces an interesting experiment transferring probes from base models to aligned models, providing direct evidence that representations of refused knowledge persist through instruction-tuning
2. The study goes beyond just finding representations; it connects them to generative behavior by showing that probe predictions correlate with the model's judgments in a separate pairwise comparison task

**Weaknesses:**

### 1. Model
The paper evaluates only small models (2B, 6B, 9B) from just two model families (Gemma and Yi). This is insufficient to support the broad claims about aligned language models in general.
Missing:
1) Evaluation on larger, more capable models (13B, 30B, 70B+)
2) Testing on diverse model families (Llama, Qwen, Mistral, etc.)
3) Discussion of why only these specific models were chosen
4) Analysis of whether findings scale with model size—do larger models show stronger or weaker linear decodability?

### 2. Data
Entity Selection: The paper provides no rationale for why "entity" is the appropriate level of analysis or why these four specific entity types (Countries, Occupations, Political Figures, Synthetic Names) were selected.
#### Missing Justification:
* Why entities rather than other knowledge types (concepts, relationships, procedures)?
* Why these four entity types among countless possibilities?
* Are these cherry-picked to show positive results, or are they representative?
* How do results vary across different knowledge domains?
#### Dataset details:
* Are all entity types mixed during training/testing or separated?
* Can probes trained on one set of entities generalize to completely new entities? This is crucial for understanding whether probes learn entity-specific patterns or general principles.
* What is the train/test split methodology? Also, the total number of data points for train/test is not specified
#### Prompt:
* Which model generates the AIM prompts?
* Were multiple generations attempted per entity? If so, how were they selected?
* Are there observable patterns in phrasing or structure that affect probe performance?
* How sensitive are results to prompt variation?
#### Confusing:
* Lines 193-199 are unclear to explain the data collection process.
### 3. Metric
* Why is Pearson correlation the primary metric rather than standard probe Mean Squared Error (MSE)?
* What does a strong negative correlation imply in Figure 3? This requires explicit explanation in the text.
* While the paper implies it, the main methodology (Section 3) should state with perfect clarity that the two variables being correlated are. In line 207, "To evaluate probe performance, we report the best layer Pearson correlation between predictions and jailbroken responses on a held-out test set". Please be specific on this 'prediction' and 'jailbroken responses'.

**Questions:**

1. How do findings scale to larger models (30B+)? Are patterns consistent or do they change?
2. Why were these specific entity types chosen? Are results consistent across other knowledge domains?
3. Can probes generalize to completely unseen entities (e.g., trained on 100 countries, tested on 50 held-out countries)?
4. What does negative correlation in Figure 3 represent?
5. Which model generates jailbreak prompts, and how sensitive are results to prompt variation?
6. What proportion of refused knowledge is linearly decodable vs. not? Are there systematic patterns in what can vs. cannot be decoded?
7. Have you attempted non-linear probing? How much additional information can be extracted?
8. Why is Pearson correlation preferred over MSE?

---

### Note · Authors · 2025-11-13

I have read and agree with the venue's withdrawal policy on behalf of myself and my co-authors.